# Biomechanical Evaluation of a Novel Non-Engaging Abutment and Screw in Internal Implant Systems: Comparative Fatigue and Load Testing

**DOI:** 10.3390/jfb16030107

**Published:** 2025-03-19

**Authors:** Su-Min Cho, Soo-Hwan Byun, So-Yee Ahn, Hyun-Sook Han, Sung-Woon On, Sang-Yoon Park, Sang-Min Yi, In-Young Park, Byoung-Eun Yang, Lee-Kyoung Kim

**Affiliations:** 1Department of Prosthodontics, Hallym University Sacred Heart Hospital, Anyang 14068, Republic of Korea; po04381@naver.com (S.-M.C.); ahnsoi0531@gmail.com (S.-Y.A.); hshan@hallym.or.kr (H.-S.H.); 2Department of Artificial Intelligence and Robotics in Dentistry, Graduate School of Clinical Dentistry, Hallym University, Chuncheon 24252, Republic of Korea; purheit@daum.net (S.-H.B.); drummer0908@hallym.or.kr (S.-W.O.); psypjy0112@naver.com (S.-Y.P.); queen21c@hallym.or.kr (S.-M.Y.); denti2875@hallym.or.kr (I.-Y.P.); 3Institute of Clinical Dentistry, Hallym University, Chuncheon 24252, Republic of Korea; 4Dental Artificial Intelligence and Robotics R&D Center, Hallym University Medical Center, Anyang 14066, Republic of Korea; 5Department of Oral and Maxillofacial Surgery, Hallym University Sacred Heart Hospital, Anyang 14066, Republic of Korea; 6Division of Oral and Maxillofacial Surgery, Hallym University Dongtan Sacred Heart Hospital, Hwaseong 18450, Republic of Korea

**Keywords:** dental implants, non-engaging abutments, implant prosthodontics, fatigue testing, biomechanical stability, combined screw- and cement-retained prostheses, implant-abutment connection, ISO 14801, prosthetic complications

## Abstract

Dental implants rely on precise prosthetic design and biomechanical stability to ensure long-term success. This study evaluates the mechanical performance of non-engaging abutments in multi-unit combined screw- and cement-retained prostheses (CSCRP) using two internal implant systems: the BlueDiamond (BD) and AnyOne (AO) systems. Unlike conventional implant systems that utilize the same type of screw for both engaging and non-engaging abutments, the BD system employs a distinct screw design for non-engaging abutments. A total of 80 implants were tested, with 40 in each group. Mechanical testing included static compressive load and fatigue tests following ISO 14801 standards. The BD system demonstrated significantly higher compressive strength (326.32 kgf vs. 231.82 kgf, *p* < 0.001) and 23.4% greater fatigue strength compared to the AO system. Precision fit analysis confirmed no significant deformation, microcracks, or fractures after 5 million loading cycles. These findings suggest that the BD system’s unique screw design for non-engaging abutments contributes to improved mechanical performance and durability. Further clinical studies are needed to assess the long-term implications of this design on prosthetic stability and implant longevity.

## 1. Introduction

Dental implants play a crucial role in oral rehabilitation, restoring function in patients with partial or complete tooth loss [1]. Their long-term success depends on osseointegration, which requires sufficient initial (primary) stability upon placement and long-term (secondary) stability through bone remodeling [2]. However, excessive biomechanical stress can lead to bone loss, implant failure, and prosthetic complications [3,4]. Optimizing implant placement and prosthetic design to distribute stress effectively is crucial for improving clinical outcomes.

Fixed dental prostheses (FDPs) are widely used in implant-supported restorations, with splinting strategies influencing load distribution and long-term stability [5]. Splinting can be performed at the implant or abutment level, each method having distinct biomechanical implications. Implant-level splinting, where multiple implants are rigidly connected, improves precision but may present challenges when implants are angulated or deeply placed [6,7]. Despite extensive research, the optimal splinting strategy remains debated [8,9].

Biomechanical studies suggest that splinting reduces stress concentration, particularly under non-axial forces. Finite element analysis and clinical studies report lower fracture and screw loosening rates in splinted restorations compared to non-splinted ones [10,11,12,13]. However, non-splinted designs also demonstrate acceptable clinical performance [14], highlighting the need for further investigation.

Technological advancements have increased the preference for implant-level restorations due to digital workflows and the advantages of screw-retained prostheses. These offer easier retrievability and a lower peri-implant disease risk than cement-retained prostheses [15,16]. A newer hybrid approach, the combined screw- and cement-retained prosthesis (CSCRP), integrates the benefits of both systems, allowing for retrievability while ensuring a precise fit [17,18,19]. In this design, a cemented suprastructure is secured over abutments while remaining retrievable through strategically placed screw access holes (SAHs) [19,20]. CSCRPs were initially developed for external hex implant connections, which allow greater angulation correction (up to 30°), but internal connections have gained popularity for their superior biomechanical stability [20,21,22,23]. One challenge in splinted CSCRP restorations is the potential difficulty in retrieval, especially when engaging abutments are used. Engaging abutments lock into the implant’s anti-rotational feature, whereas non-engaging abutments lack this connection, permitting rotational freedom and improving retrievability in multi-unit restorations [24,25]. However, their mechanical performance under functional loading conditions remains underexplored [26].

This study evaluates the mechanical performance of newly designed non-engaging abutments in multi-unit CSCRP restorations. The BD system compensates for angulation by shortening the abutment length entering the fixture and increasing the length and thickness of the screw without sacrificing strength (Figure 1 and Figure 2). A comparative fatigue test was conducted with two internal implant systems—BlueDiamond (BD, MegaGen Implant Co., Daegu, Republic of Korea) and AnyOne (AO, MegaGen Implant Co., Daegu, Republic of Korea)—featuring non-engaging abutments with 15-degree and 11-degree internal tapers, respectively (Figure 1). The primary objective was to assess the clinical applicability of the BD system in multi-unit restorations. The null hypothesis was that the BD system would demonstrate non-inferiority in fatigue strength compared to the AO system under dynamic loading conditions.

## 2. Materials and Methods

### 2.1. Study Design and Implant Groups

This study was designed as a comparative analysis of two implant systems, employing a total of 80 dental implants evenly divided into two groups: the experimental (BD) group and the control (AO) group. Each group comprised 40 implants, with 20 two-unit prostheses fabricated per group. The implant sites were standardized based on pre-planned prosthetic dimensions for the first molars and second premolars.

In the BD group, Ø4.4 mm × 8 mm (ARO4408) implants were used for the second premolar region, and Ø4.8 mm × 8 mm (ARO4808) implants for the first molar region (Figure 3, Table 1). The AO group received Ø4.5 mm × 8 mm (IF4508) implants for the second premolar and Ø5.0 mm × 8 mm (IF5008) implants for the first molar (Figure 4, Table 1). The selection of implant dimensions and abutment types was based on prior literature recommendations [27,28,29]. Current research suggests that implant diameters typically range from 3.5 mm to 4.5 mm for second premolars and from 4.5 mm to 5.7 mm for first molars [27]. Both groups employed non-engaging abutments: the BD group used non-engaging (non-octa) abutments, while the AO group used non-engaging (non-hex) abutments (Figure 5).

### 2.2. Prosthesis Fabrication and Assembly

Implant fixtures and abutments were assembled and torqued according to the manufacturers’ recommended values using a calibrated torque meter. To ensure reproducibility, all specimens were mounted in a custom-designed jig. The prostheses were fabricated using a standardized workflow: impressions were taken at the abutment level, wax patterns were created, and non-precious metal alloy restorations were cast. Each group included twenty two-unit prostheses, corresponding to 40 implants per group. Cementation was performed using RelyX Unicem (3M ESPE, St. Paul, MN, USA) under a standardized load for 4 min to ensure uniform bonding.

### 2.3. Mechanical Testing Protocol

Mechanical testing followed ISO 14801 fatigue testing standards [30] for dental implants. A custom testing jig was designed to apply loading at a 30° angle to the long axis of the implant–abutment complex.

#### 2.3.1. Compressive Load Test

The maximum compressive strength was determined using an Instron 3366 universal testing machine (Instron Co., Norwood, MA, USA). A 30° inclined load was applied at a fixed 11 mm distance from the fixture at a loading rate of 1.0 mm/min (Figure 6). Load-strain curves were recorded until plastic deformation occurred, and the maximum compressive strength was extracted. Ten two-unit cap specimens were tested per group.

#### 2.3.2. Fatigue Test

Fatigue testing used the same specimen preparation and fixture protocols as the compressive load test. An Instron ElectroPuls E3000 fatigue tester (Instron Co., Norwood, MA, USA) applied 5 million cyclic loads at 2–15 Hz under 20 ± 5 °C conditions. The initial fatigue load was set at 80% of the average compressive strength. If a specimen fractured before completing 5.0 × 10^6^ cycles, the fatigue load was reduced incrementally to 80% of the previous load until the fatigue limit load was determined. An additional refinement test increased the fatigue limit load by 10% to establish a more precise threshold.

#### 2.3.3. Precision Fit and Surface Analysis

Following fatigue testing, precision fit analysis assessed fastening accuracy, deformation, cracking, and fracture. Intact specimens were embedded in acrylic resin, heated for 10 min at 165 °C, and cooled for 7 min using an automatic mounting press (HM-130, Korea-Tech, Yangsan-si, Republic of Korea). The implant–abutment interface was cross-sectioned using an automatic precision cutting machine (Minitom, Struers, Copenhagen, Denmark) at 400 rpm, and then polished using an automated polisher (P2-250B + AH32, Korea-Tech, Yangsan-si, Republic of Korea). To enhance conductivity and surface visualization, a gold-ion coating was applied (G20, GSEM Co., Suwon-si, Republic of Korea). A scanning electron microscope (SEM; AIS1800C, Serontech, Uiwang-si, Republic of Korea) evaluated contact gaps, microcracks, and structural deformations.

### 2.4. Statistical Analysis

All statistical analyses were performed using SPSS software Version 22.0 (SPSS Inc., Chicago, IL, USA). To compare maximum strength values between the BD and AO groups, the Mann–Whitney U test was applied. A statistical significance level of α = 0.05 was used.

## 3. Results

### 3.1. Static Compression-Strength Test Results

The results of the compressive load test for the BD and AO implant groups are summarized in Table 2 (Figure 7). The BD group exhibited a significantly higher mean maximum compressive strength than the AO group. Specifically, the BD implants demonstrated a mean compressive strength of 326.32 ± 21.09 kgf, whereas the AO implants exhibited a mean compressive strength of 231.82 ± 11.33 kgf. Statistical analysis confirmed a significant difference between the two groups (t = 8.87, *p* < 0.001), indicating that the BD implant system possesses superior compressive strength compared to the AO implant system. These findings suggest that BD implants may better withstand occlusal forces in clinical applications.

### 3.2. Fatigue Test Results

The results of the fatigue test demonstrated that BD implants exhibited a 23.4% higher fatigue strength than AO implants. The maximum fatigue load was 1141.94 N for the BD group and 925.61 N for the AO group (Figure 8, Table 3). These findings indicate that BD implants withstand higher cyclic loading conditions before failure, suggesting enhanced durability under functional loading. The increased fatigue resistance of BD implants may contribute to long-term stability and implant longevity.

### 3.3. Precision Fit and Structural Integrity

Following 5 million fatigue cycles, the precision fit test revealed that the implant–abutment interface gap remained below 10 μm in both groups, indicating no significant changes in fit accuracy (Figure 9). The number of bridges used to obtain the fatigue load values was 14 for the BD group and 11 for the AO group. Importantly, no fracture, cracking or deformation was observed in any specimen at their respective fatigue load values, suggesting that both implant systems maintained structural integrity under prolonged cyclic loading.

## 4. Discussion

This study demonstrated that the BD implant system exhibits superior mechanical performance compared to the AO system, as evidenced by its higher compressive strength and fatigue resistance. Static compression testing revealed that BD implants withstood significantly higher maximum compressive loads (*p* = 0.008), while fatigue testing indicated a 23.4% increase in fatigue strength for BD implants compared to AO implants. Additionally, precision fit analysis showed that the implant–abutment interface gap remained below 10 μm in both groups, with no observed fractures, cracking, or deformation after 5 million cycles. These findings suggest that BD implants offer mechanical advantages, potentially reducing the risk of implant failure and prosthetic complications.

Fatigue testing is widely recognized as one of the most clinically relevant methods for assessing the long-term performance of dental implants [31]. The 30° loading angle used in this study aligns with ISO 14801:2016 standards [30] and is consistent with previous in vitro studies on implant biomechanics [32,33]. The 5-million-cycle fatigue test assumes that an implant functions for at least 5 to 10 years, based on an individual’s masticatory habits—three meals per day at a rate of 1 to 2 Hz, lasting 3 to 15 min per meal [34,35]. However, because dental implant fatigue testing involves significantly higher repetitions than regular masticatory forces, an implant that withstands 5 million cycles at the fatigue limit load is likely to have a real-world lifespan exceeding 10 years.

Since the introduction of osseointegrated dental implants by Brånemark in the 1960s [36], advancements in implant design, surface treatment, and biomechanics have significantly improved early implant success rates. However, late failure remains a concern, primarily due to prosthetic complications, implant fractures, and loss of osseointegration [37]. This study highlights the importance of prosthetic design and load distribution in mitigating these risks. One of the key biomechanical challenges in multi-unit restorations is achieving passive fit. Engaging abutments concentrate rotational and lateral forces on the implant, increasing the risk of screw loosening, fracture, or stress concentration [10]. In contrast, non-engaging abutments enhance passive fit and allow for greater flexibility in multi-unit restorations, thereby reducing mechanical complications.

The final implant prosthesis can be categorized into screw-retained prostheses (SRPs), cement-retained prostheses (CRPs), and combined screw- and cement-retained prostheses (CSCRPs), each with distinct advantages and limitations [17,38,39]. SRPs offer superior retrievability, facilitating the management of screw loosening or prosthetic fractures. Additionally, they eliminate the risks associated with excess cement, which can contribute to peri-implantitis. However, SRPs may pose esthetic and mechanical challenges due to screw access holes (SAHs) and occlusal stress-related screw loosening or fractures [40,41]. CRPs improve esthetics by eliminating SAHs and ensuring perpendicular occlusal force distribution. However, retrievability is limited, complicating repairs, and residual cement at the gingival margin may increase peri-implantitis risk [40,41]. CSCRPs address the limitations of SRPs and CRPs [17,42]. CSCRPs improve retrievability while maintaining a passive fit, reducing misfit-related complications [43,44]. Achieving superior passive fit minimizes stress on the implant and strain on the prosthesis [45]. Additionally, the SAH in CSCRP allows for cement removal before final curing, lowering the risk of peri-implantitis while ensuring easier retrieval [15,17,20,42]. However, SAH positioning must be carefully managed to prevent improper occlusal placement [21,46,47], a challenge that can be mitigated with angulated screw channel technology [48,49].

This study applied the CSCRP methodology to internal connection implants, which offer biomechanical advantages over external connections [22,23,24]. The AO system shares a connection angle with Astra implants (11-degree taper) (Astra Tech AB, Mölndal, Sweden), while the BD system aligns with BLT implants (15-degree taper) (Institut Straumann AG, Basel, Switzerland), enhancing abutment compatibility and reinforcing the clinical applicability of BD implants. In multi-unit restorations, engaging abutments may complicate retrievability due to fixture misalignment. BD implants feature an extended fixture–abutment connection (4.35 mm) compared to AO implants (2.64 mm) (Figure 1), which may introduce challenges in CSCRP applications. To address this, non-engaging abutments were introduced to improve passive fit and angulation compensation. The EZ post non-engaging abutment system reduces the connection length to 1.35 mm, improving retrievability and load distribution (Figure 1 and Figure 2).

The BD system incorporates several structural advancements to optimize implant mechanics. By increasing abutment screw length and thickness, BD implants compensate for stress concentration due to connection geometry changes (Figure 1). The BD non-engaging abutment screws measure 2.4 mm in diameter—twice the thickness of engaging abutment screws—allowing engagement with non-engaging abutments while maintaining connection strength (Figure 1). Previous studies indicate that larger abutment screw diameters correlate with greater load resistance [50], suggesting that BD screws experience lower stress under various loading conditions, thereby reducing fracture risk. Finite element analysis of the EZ post-non-engaging abutment implementation demonstrated that peak von Mises stress at the abutment–screw interface was three times lower in BD implants than in AO implants [51]. Furthermore, maximum principal stress in the upper prosthesis was also lower in BD implants, aligning with this study’s findings of higher compressive and fatigue strengths. These results indicate that BD implants, through optimized screw geometry and load distribution, exhibit greater resistance to mechanical failure, contributing to long-term clinical success. Another advantage of fabricating fixed partial dentures (FPDs) using BD fixtures and non-engaging abutments is that the abutments can be retrieved along with the upper prosthesis, even when they are not perfectly parallel, with a theoretical maximum divergence of up to 30 degrees. In contrast, FPDs utilizing the AO system can only be retrieved at a maximum divergence of 22 degrees.

The two groups of upper-prosthesis fabrication methods used in this study were CSCRPs and implant-level restoration. The implant-level restoration was selected due to its advantages in retrievability and maintenance, allowing for easier repairs, hygiene maintenance, and implant monitoring, which reduces long-term complications [52]. Digital workflow integration is more efficient with implant-level scanning, eliminating the need for physical impressions and improving workflow precision [53]. The peri-implantitis risk is minimized, as cement-retained restorations have been associated with residual cement leading to inflammation. The shift toward the use of CSCRPs or SRPs reduces this risk [54]. Multi-unit abutment advancements have made implant-level restorations more viable, even in cases where implant angulation discrepancies exist [55]. Cost-effectiveness is also improved with implant-level restoration, as no additional abutments are required. One study showed that abutments could counterbalance potential misfits between the framework and implant fixture in the abutment-level group, whereas in the implant-level group, potential misfits could cause uneven stresses and strains between the framework and implant fixture [9]. However, with the CSCRP approach, the cement space can offset potential mismatches even in the implant-level approach.

Despite its strengths, this study has several limitations. Future research should incorporate anatomically contoured prostheses with SAH placement to better simulate real-world conditions. Larger sample sizes are needed for parametric statistical analysis and normality assessments. Additional testing is required to assess long-term stability, including wear and corrosion testing, thermo-mechanical fatigue testing, analysis of micromovement at the implant–abutment interface, and assessment of bacterial microleakage. Clinical trials should be conducted to evaluate long-term outcomes in diverse patient populations under various loading conditions. Addressing these limitations will enhance the generalizability of these findings and further validate the clinical applicability of BD implants.

## 5. Conclusions

This study demonstrated that the BD group exhibits superior mechanical performance compared to the AO group, as evidenced by its higher compressive and fatigue strength. These findings suggest that BD implants may serve as a clinically viable option for multi-unit restorations, offering an alternative to AO implants. However, the shortened and non-engaging abutment design, along with its corresponding screw, introduces potential risks, including mechanical complications, which necessitate careful clinical monitoring. Further long-term clinical studies are required to evaluate the impact of these design modifications on implant longevity, functional stability, and prosthetic process.

## Figures and Tables

**Figure 1 jfb-16-00107-f001:**
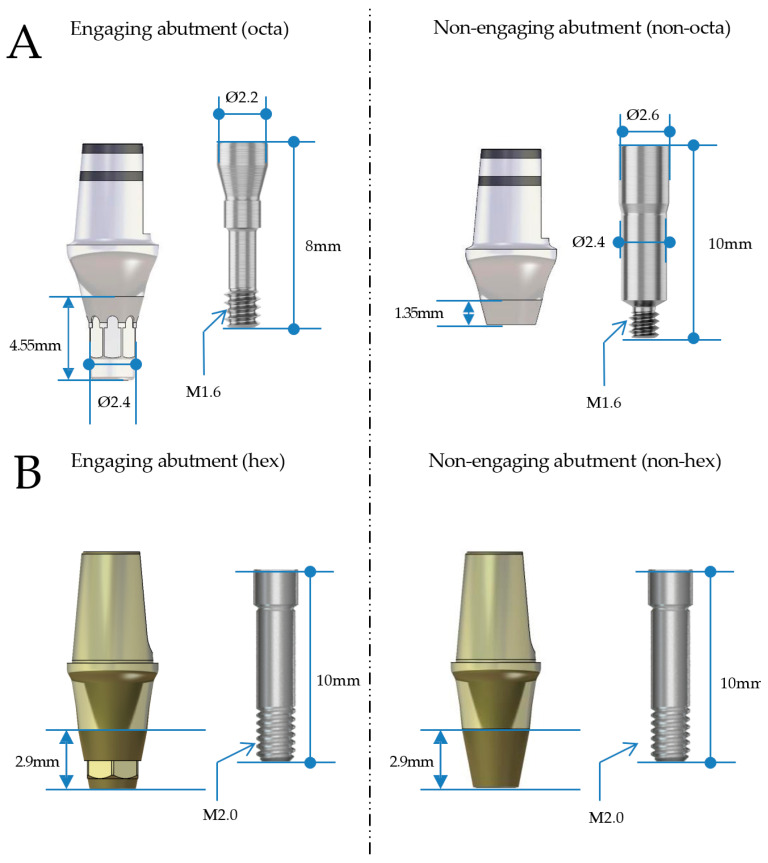
(**A**) Differences between engaging and non-engaging abutments in the BlueDiamond (BD) system. (**B**) Differences between engaging and non-engaging abutments in the AnyOne (AO) system.

**Figure 2 jfb-16-00107-f002:**
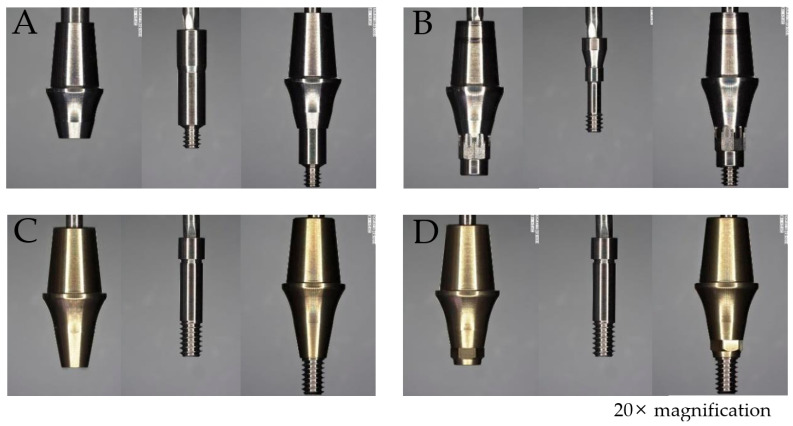
Photographs of abutments and screws under 20× magnification, showing individual components (left to right: abutment, screw, and assembled abutment-screw connection). (**A**) Non-engaging abutment of the BD system. (**B**) Engaging abutment of the BD system. (**C**) Non-engaging abutment of the AO system. (**D**) Engaging abutment of the AO system. Notably, the BD system utilizes different screw designs for engaging and non-engaging abutments (**A**,**B**), whereas the AO system uses the same screw for both abutment configurations (**C**,**D**).

**Figure 3 jfb-16-00107-f003:**
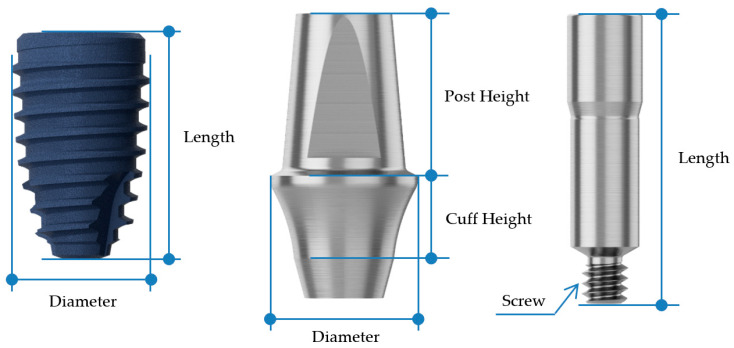
BD fixture, non-engaging abutment, and corresponding abutment screw used in this study.

**Figure 4 jfb-16-00107-f004:**
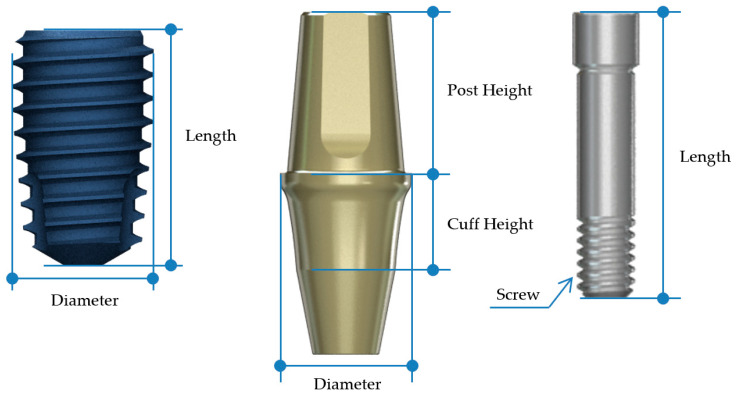
AO fixture, non-engaging abutment, and corresponding abutment screw used in this study.

**Figure 5 jfb-16-00107-f005:**
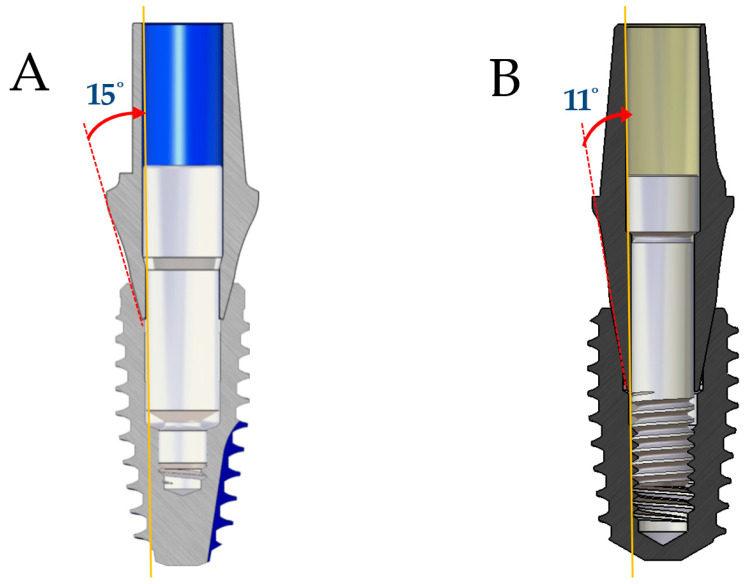
(**A**) BD fixture with corresponding non-engaging abutment. (**B**) AO fixture with corresponding non-engaging abutment.

**Figure 6 jfb-16-00107-f006:**
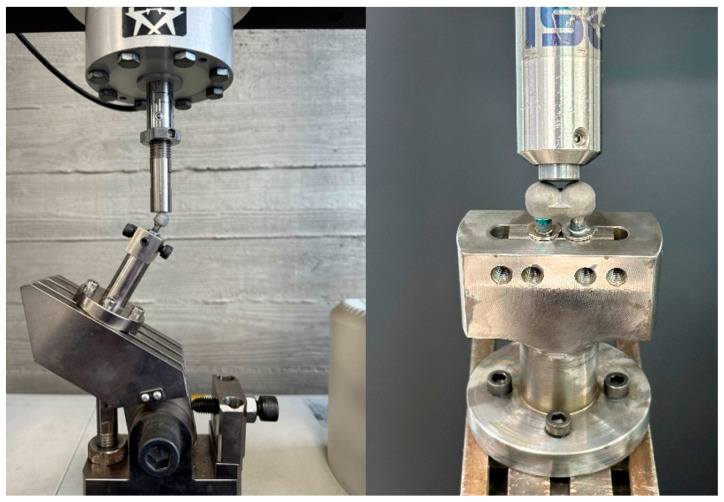
Sample preparation and positioning for the compressive load test. A 30° angled load was applied until plastic deformation occurred. Both crowns were loaded simultaneously.

**Figure 7 jfb-16-00107-f007:**
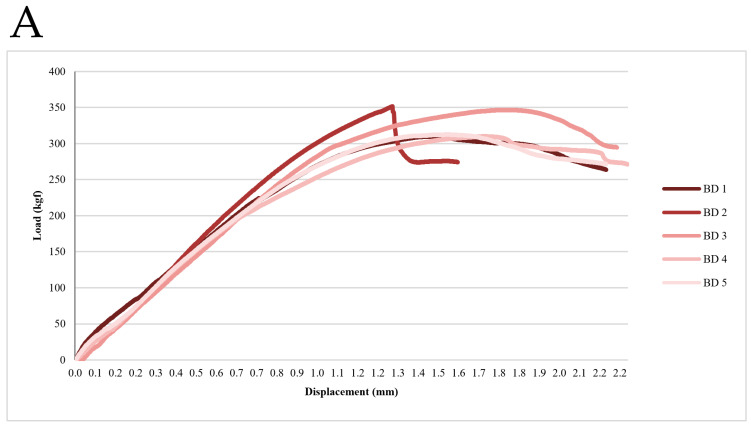
Load–displacement curves from the static compression test. These curves were used to determine the maximum compressive strength. (**A**) BD group. (**B**) AO group.

**Figure 8 jfb-16-00107-f008:**
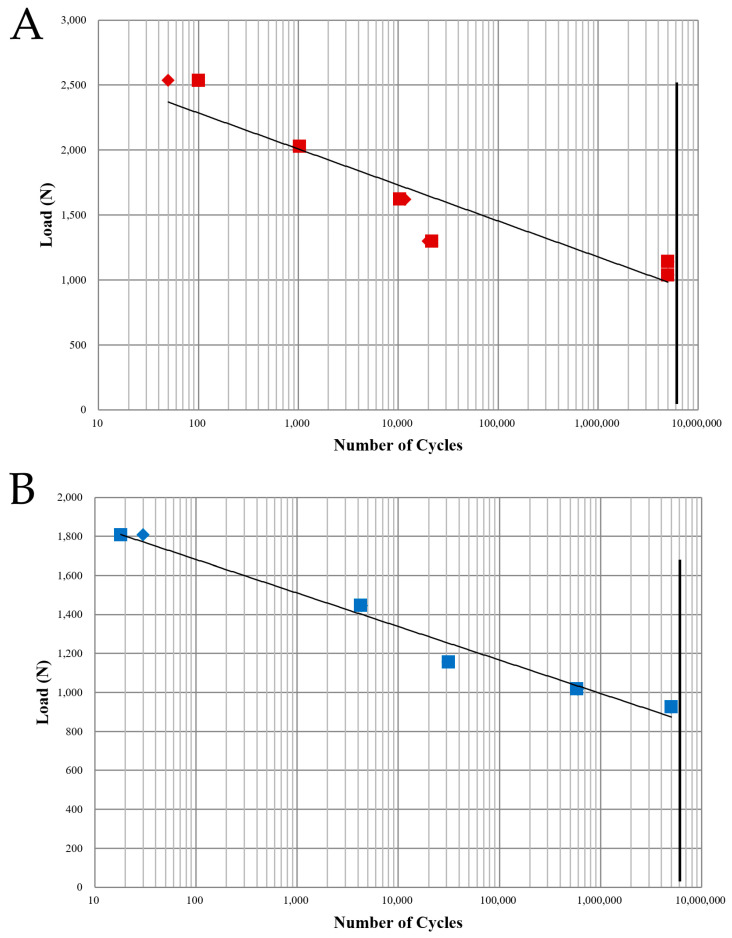
Load–cycle curves from the fatigue test. The initial fatigue load was set at 80% of the average compressive strength. If a specimen fractured before completing 5.0 × 10^6^ cycles, the fatigue load was reduced to 80% of the previous load until the fatigue limit load was determined. After load determination, two additional specimens were tested for a total of three per group. The bold black line is the trendline plotted by the logarithmic model. (**A**) BD group. (**B**) AO group.

**Figure 9 jfb-16-00107-f009:**
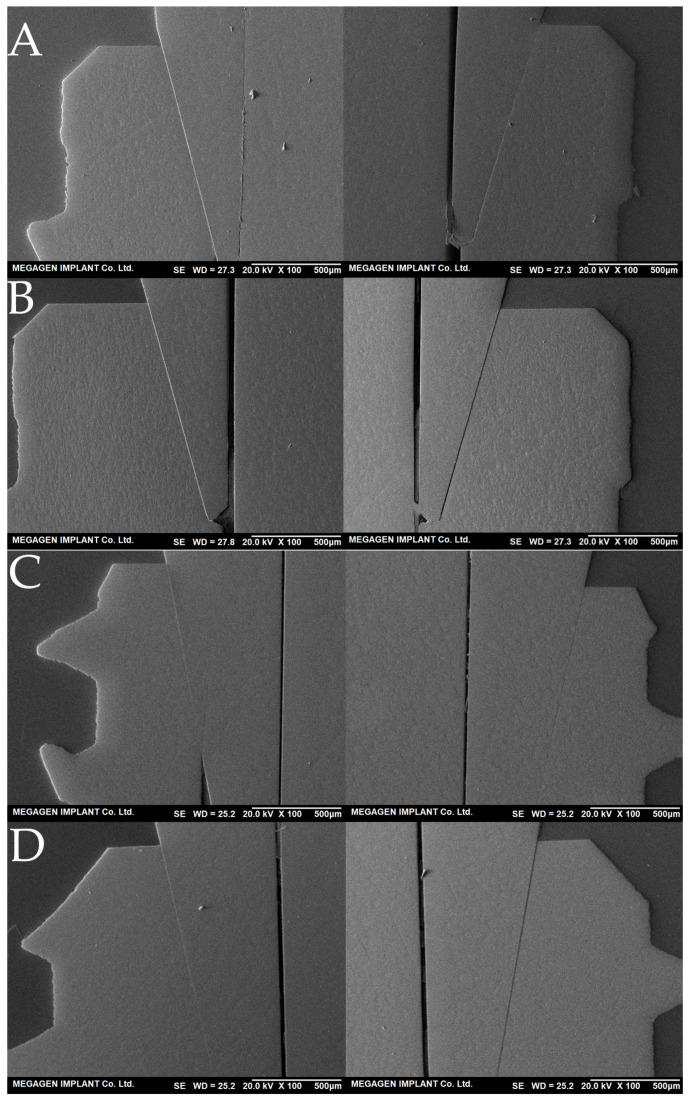
Scanning electron microscope (SEM) images of precision fit analysis following the fatigue test. Cross-sectional images of the implant–abutment interface reveal that all gap measurements were ≤10 μm. (**A**) Second premolar region of the BD group. (**B**) First molar region of the BD group. (**C**) Second premolar region of the AO group. (**D**) First molar region of the AO group.

**Table 1 jfb-16-00107-t001:** Specifications of the implant fixture, abutment and abutment screw used in this study.

		Fixture		Abutment			Abutment Screw	
		Diameter	Length (mm)	Diameter	Cuff Height (mm)	Post Height (mm)	Screw	Length (mm)
BD	2nd premolar	Ø4.4	8	Ø5.0	3	5.5	M1.6	10
	1st molar	Ø4.8	8	Ø7.0	3	5.5		
AO	2nd premolar	Ø4.5	8	Ø4.5	3	5.5	M2.0	10
	1st molar	Ø5.0	8	Ø6.5	3	5.5		

**Table 2 jfb-16-00107-t002:** Maximum compressive strength.

	BD (kgf)	AO (kgf)
1	309.98	218.41
2	351.75	238.66
3	346.87	225.15
4	310.25	227.96
5	312.75	246.92
mean ± std	326.32 ± 21.09	231.82 ± 11.33
t-value	8.88	
*p*-value	<0.001	

**Table 3 jfb-16-00107-t003:** Maximum fatigue load.

	BD					
	1st	2nd	3rd	4th	5th	6th
Load (N)	2534.59	2027.67	1622.1	1297.72	1038.21	1141.94
Specimen 1 (cycles)	50	997	11,580	19,849	5.0 × 10^6^	5.0 × 10^6^
Specimen 2 (cycles)	100	1028	10,367	21,733	5.0 × 10^6^	5.0 × 10^6^
Specimen 3 (cycles)	-	-	-	-	5.0 × 10^6^	5.0 × 10^6^
	AO					
	1st	2nd	3rd	4th	5th	6th
Load (N)	1807.83	1446.26	1156.99	925.61	1018.12	-
Specimen 1 (cycles)	30	4304	31,081	5.0 × 10^6^	590,184	-
Specimen 2 (cycles)	18	4268	31,570	5.0 × 10^6^	583,092	-
Specimen 3 (cycles)	-	-	-	5.0 × 10^6^	-	-

## Data Availability

The original contributions presented in the study are included in the article, further inquiries can be directed to the corresponding author.

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
