# Peer review of "Biomechanical Evaluation of a Novel Non-Engaging Abutment and Screw in Internal Implant Systems: Comparative Fatigue and Load Testing"

_jfb, 2025, doi:10.3390/jfb16030107_

Round 1

Reviewer 1 Report

Comments and Suggestions for Authors

Dear Authors,

I would like to commend you on your manuscript titled "Biomechanical Evaluation of a Novel Non-Engaging Abutment and Screw in Internal Implant Systems: Comparative Fatigue and Load Testing." Your work addresses an important topic in the field of implantology, and it is clear that you have put considerable effort and attention into your research. However, I would have a few suggestions:

Introduction section is too long. Try to introduce readers a little more directly and purposefully to what will be the subject of your evaluation.

Line 85 . superstructure or suprastructure?

Line 132. it is not clear how there were 40 samples (implants) in each group?

Line 134-138. If the position of implant „second premolar region “, „first molar region“ refers to the position in jig please state so. If that is the case, you should add short explanation about this jig. In the jig presented at Fig 6 it is not clear how it can be distinguished which tooth position is loaded?

In line 156 you first time state „To ensure reproducibility, all specimens were mounted in a custom-designed jig “ This, with more detailed description should be stated at the beginning of M&M section.

Add sample numbers at the beginning of the M&M section. Are the numbers in Tables 2 and 3 actually individual samples? This needs explanation.

Which device was used to make cross sections for SEM evaluation?

Reviewer 2 Report

Comments and Suggestions for Authors

March, 3, 2025

Dear authors

Thank you for an interesting report.

You investigated the mechanical performance of non-engaging abutments in two internal implant systems, BlueDiamond (BD) and AnyOne (AO), through static compressive load and fatigue testing following ISO 14801 standards. The results indicate that the BD system’s unique screw design enhances mechanical strength and durability. Given its focus on dental implant stability, this research is particularly relevant to dental professionals. As such, I believe this study is clinically significant. I also think that this is a suitable article for the journal that will be of interest to the readers, particularly those involved in prosthetic treatment. I acknowledge the validity of several of your claims; however, I think that several revisions are required as follows:   

1.Introduction

  1. Since the journal's readers come from a wide range of research fields, likely, researchers studying implants in orthopedics and general surgery (including orthopedics and maxillofacial surgery) will likely find the dental-specific technical terms difficult to understand. At the very least, I think you should consider the ease of reading for readers who do not have deep specialized knowledge of implantology, even in the dental field. Therefore, I recommend adding explanations for overly technical terms, rather than general dental terms, to make the paper easier to understand.

For example, “non-engaging abutment”, “primary and secondary stabilization (with clear definitions)”, “implant-level/ abutment level splinting”, etc.

2.Materials and Methods

No major concerns in this section

3.Results

1) Table 1 and Page 8. L204-205

The mean values are reported with two decimal places, while the standard deviation values are presented with three decimal places. To maintain consistency and ensure proper numerical representation, the standard deviation values should be rounded to match the decimal places of the mean values. Therefore, I recommend adjusting the standard deviation values accordingly to improve clarity and precision in data presentation.

 2) Figure 7

The physical properties represented on the vertical and horizontal axes are not explicitly labeled. To ensure clarity and facilitate proper interpretation of the data, I recommend adding appropriate labels to both axes. This addition could improve readability and allow readers to understand the presented results more accurately.

 3) Figure 8

The thick vertical line in Figure 8 lacks explanation. I recommend adding an illustration of what it means.

4. Discussion

 1) Page 11 line 250

In the sentence "The static compression test confirmed that BD implants withstood significantly higher maximum compressive loads (P = .008)", it appears that the P value refers to the statistical p-value. However, this notation is inconsistent with other parts of the manuscript, where lowercase p is used (e.g., p < 0.001). To maintain consistency and adhere to standard formatting, I recommend changing "P = .008" to "p = 0.008".

 2) Page 11 lines 262-264

In the sentence "However, late failure remains a concern, primarily due to prosthetic complications, implant fractures, and loss of osseointegration," no references are provided to support these claims. Since these factors are critical in understanding implant longevity, I think you should cite relevant literature that discusses the primary causes of late implant failure.

 3) Page 12 lines 256-259

The fatigue test follows ISO 14801:2016 standards, which ensures methodological rigor. However, it would enhance clarity if you additionally explained the interpretation of the clinical significance of the 5-million-cycle fatigue test. For instance, estimating how many years of typical intraoral function this corresponds to would help contextualize the results for the readers.

 4) Page 13 lines 341-347

This study evaluates the mechanical strength of implant abutments, providing valuable insights for clinical applications. However, to comprehensively assess the long-term stability of implants, additional tests should be conducted. In particular, considering the potential impact of differences between non-engaging and engaging abutments, the following investigations are recommended.

  1 Wear and Corrosion Testing

 2 Thermo-Mechanical Fatigue Testing

 3 Micro-Movement Analysis of the Implant-Abutment Connection

 4 Bacterial Microleakage Evaluation

  1. Conclusions

No major concerns in this section.

Round 2

Reviewer 2 Report

Comments and Suggestions for Authors

March, 15, 2025

Dear authors

Thank you for providing an interesting and engaging report.

I have reviewed again the revised manuscript and confirmed that almost all the points raised in my previous peer review have been addressed. It is evident that you have made a significant effort to revise the paper, and in my opinion, the quality of the manuscript has been greatly improved. I have no additional comments or concerns to raise in this review.

Best regards,